# Revisiting Loss Modelling for Unstructured Pruning

## Abstract

By removing parameters from deep neural networks, unstructured pruning methods aim at cutting down memory footprint and computational cost, while maintaining prediction accuracy. In order to tackle this otherwise intractable problem, many of these methods model the loss landscape using first or second order Taylor expansions to identify which parameters can be discarded. We revisit loss modelling for unstructured pruning: we show the importance of ensuring locality of the pruning steps, and systematically compare first and second order Taylor expansions. Finally, we show that better preserving the original network function does not necessarily transfer to better performing networks after fine-tuning, suggesting that only considering the impact of pruning on the loss might not be a sufficient objective to design good pruning criteria.

## 1 Introduction

Neural networks are getting bigger, requiring more and more computational resources not only for training, but also when used for inference. However, resources are sometimes limited, especially on mobile devices and low-power chips. In unstructured pruning, the goal is to remove some parameters (i.e. setting them to zeros), while still maintaining good prediction performances. This is fundamentally a combinatorial optimization problem which is intractable even for small scale neural networks, and thus various heuristics have been developed to prune the model either before training (Lee et al., 2019b; Wang et al., 2020), during training (Louizos et al., 2017; Molchanov et al., 2017; Ding et al., 2019), or in an iterative training/fine-tuning fashion (LeCun et al., 1990; Hassibi & Stork, 1993; Han et al., 2015; Frankle & Carbin, 2018; Renda et al., 2020).

Early pruning work Optimal Brain Damage (OBD) (LeCun et al., 1990), and later Optimal Brain Surgeon (OBS) (Hassibi & Stork, 1993), proposed to estimate the importance of each parameter by approximating the effect of removing it, using the second order term of a Taylor expansion of the loss function around converged parameters. This type of approach involves computing the Hessian, which is challenging to compute since it scales quadratically with the number of parameters in the network. Several approximations have thus been explored in the literature (LeCun et al., 1990; Hassibi & Stork, 1993; Heskes, 2000; Zeng & Urtasun, 2019; Wang et al., 2019). However, state-of-the-art unstructured pruning methods typically rely on Magnitude Pruning (MP) (Han et al., 2015), a simple and computationally cheap criterion based on weight magnitude, that works extremely well in practice (Renda et al., 2020).

This paper revisits linear and diagonal quadratic models of the local loss landscape for unstructured pruning. In particular, since these models are *local* approximations and thus assume that pruning steps correspond to small vectors in parameter space, we propose to investigate how this locality assumption affects their performance. Moreover, we show that the *convergence* assumption behind OBD and OBS, which is overlooked and violated in current methods, can be relaxed by maintaining the gradient term in the quadratic model. Finally, to prevent having to compute second order information, we propose to compare diagonal quadratic models to simpler linear models.

While our empirical study demonstrates that pruning criteria based on linear and quadratic loss models are good at preserving the training loss, it also shows that this benefit does not necessarily transfer to better networks after fine-tuning, suggesting that preserving the loss might not be the best objective to optimize for. Our contributions can be summarized as follows:

1. We present pruning criteria based on both linear and diagonal quadratic models of the loss, and show how they compare at preserving training loss compared to OBD and MP.
2. We study two strategies to better enforce locality in the pruning steps, pruning in several stages and regularising the step size, and show how they improve the quality of the criteria.
3. We show that using pruning criteria that are better at preserving the loss does not necessarily transfer to better fine-tuned networks, raising questions about the adequacy of such criteria.

## 2 BACKGROUND: UNSTRUCTURED PRUNING

### 2.1 UNSTRUCTURED PRUNING PROBLEM FORMULATION

For a given architecture, neural networks are a family of functions $f_{\boldsymbol{\theta}} : \mathcal{X} \to \mathcal{Y}$ from an input space $\mathcal{X}$ to an output space $\mathcal{Y}$, where $\boldsymbol{\theta} \in \mathbb{R}^D$ is the vector that contains all the parameters of the network. Neural networks are usually trained by seeking parameters $\boldsymbol{\theta}$ that minimize the empirical risk $\mathcal{L}(\boldsymbol{\theta}) = \frac{1}{N} \sum_i \ell(f_{\boldsymbol{\theta}}(x_i), t_i)$ of a loss function $\ell$ on a training dataset $\mathcal{D} = \{(x_i, t_i)\}_{1 \le i \le N}$, composed of $N$ (example, target) pairs.

The goal of unstructured pruning is to find a step $\Delta\boldsymbol{\theta}$ to add to the current parameters $\boldsymbol{\theta}$ such that $\|\boldsymbol{\theta} + \Delta\boldsymbol{\theta}\|_0 = (1 - \kappa)D$, i.e. the parameter vector after pruning is of desired sparsity $\kappa \in [0, 1]$. While doing so, the performance of the pruned network should be maintained, so $\mathcal{L}(\boldsymbol{\theta} + \Delta\boldsymbol{\theta})$ should not differ much from $\mathcal{L}(\boldsymbol{\theta})$. Unstructured pruning thus amounts to the following minimization problem:

$$\underset{\Delta\boldsymbol{\theta}}{\text{minimize}} \quad \Delta\mathcal{L}(\boldsymbol{\theta}, \Delta\boldsymbol{\theta}) \overset{\text{def}}{=} |\mathcal{L}(\boldsymbol{\theta} + \Delta\boldsymbol{\theta}) - \mathcal{L}(\boldsymbol{\theta})| \qquad \text{s.t.} \quad \|\boldsymbol{\theta} + \Delta\boldsymbol{\theta}\|_0 = (1 - \kappa)D \qquad (1)$$

Directly solving this problem would require evaluating $\mathcal{L}(\boldsymbol{\theta} + \Delta\boldsymbol{\theta})$ for all possible values of $\Delta\boldsymbol{\theta}$, which is prohibitively expensive, so one needs to rely on heuristics to find good solutions.

### 2.2 OPTIMAL BRAIN DAMAGE CRITERION

Optimal Brain Damage (OBD) (LeCun et al., 1990) proposes to use a quadratic modelling of $\mathcal{L}(\boldsymbol{\theta} + \Delta\boldsymbol{\theta})$, leading to the following approximation of $\Delta\mathcal{L}(\boldsymbol{\theta}, \Delta\boldsymbol{\theta})$:

$$\Delta\mathcal{L}^{QM}(\boldsymbol{\theta}, \Delta\boldsymbol{\theta}) = \left| \frac{\partial\mathcal{L}(\boldsymbol{\theta})}{\partial\boldsymbol{\theta}}^{\top} \Delta\boldsymbol{\theta} + \frac{1}{2}\Delta\boldsymbol{\theta}^{\top}\mathbf{H}(\boldsymbol{\theta})\Delta\boldsymbol{\theta} \right| \qquad (2)$$

where $\mathbf{H}(\boldsymbol{\theta})$ is the Hessian of $\mathcal{L}(\boldsymbol{\theta})$. $\mathbf{H}(\boldsymbol{\theta})$ being intractable, even for small-scale networks, its Generalized Gauss-Newton approximation $\mathbf{G}(\boldsymbol{\theta})$ (Schraudolph, 2002) is used in practice, as detailed in Appendix A.[1] Then, two more approximations are made: first, it assumes the training of the network has converged, thus the gradient of the loss wrt $\boldsymbol{\theta}$ is 0, which makes the linear term vanish. Then, it neglects the interactions between parameters, which corresponds to a diagonal approximation of $\mathbf{G}(\boldsymbol{\theta})$, leading to the following model:

$$\Delta\mathcal{L}^{OBD}(\boldsymbol{\theta}, \Delta\boldsymbol{\theta}_k) \approx \frac{1}{2}\mathbf{G}_{kk}(\boldsymbol{\theta})\Delta\boldsymbol{\theta}_k^2 \qquad \Rightarrow \qquad s_k^{\text{OBD}} = \frac{1}{2}\mathbf{G}_{kk}(\boldsymbol{\theta})\boldsymbol{\theta}_k^2 \qquad (3)$$

$s_k^{\text{OBD}}$ is the *saliency* of each parameter, estimating how much the loss will change if that parameter is pruned, so if $\Delta\boldsymbol{\theta}_k = -\boldsymbol{\theta}_k$. Parameters can thus be ranked by order of importance, and the ones with the smallest saliencies (i.e. the least influence on the loss) are pruned, while the ones with the biggest saliencies are kept unchanged. This can be interpreted as finding and applying a binary mask $\mathbf{m} \in \{0, 1\}^D$ to the parameters such that $\boldsymbol{\theta} + \Delta\boldsymbol{\theta} = \boldsymbol{\theta} \odot \mathbf{m}$, where $\odot$ is the element-wise product.

### 2.3 MAGNITUDE PRUNING CRITERION

Magnitude Pruning (MP) (Han et al., 2015), is a popular pruning criterion in which the saliency is simply based on the norm of the parameter:

$$s_k^{\text{MP}} = \boldsymbol{\theta}_k^2 \qquad (4)$$

Despite its simplicity, MP works extremely well in practice (Gale et al., 2019), and is used in current state-of-the-art methods (Renda et al., 2020). We use global MP as baseline in all our experiments.

---

[1]Although LeCun et al. (1990) uses $\mathbf{H}(\boldsymbol{\theta})$ in the equations of OBD, it is actually $\mathbf{G}(\boldsymbol{\theta})$ which was used in practice (LeCun, 2007).

## 2.4 Optimal Brain Surgeon

Optimal Brain Surgeon (OBS) (Hassibi & Stork, 1993) also relies on the quadratic model in Equation 2 to solve the minimization problem given in Equation 1, but uses the Lagrangian formulation to include the constraint to the solution of the minimization problem. Since OBS requires to compute the inverse of $\mathbf{H}(\boldsymbol{\theta})$, several approximations have been explored in the literature, including diagonal, as in the original OBS, Kronecker-factored (Martens & Grosse, 2015) as in ML-Prune (Zeng & Urtasun, 2019), or diagonal, but in an Kronecker-factored Eigenbasis (George et al., 2018), as in EigenDamage (Wang et al., 2019). While we use OBD in our demonstrations and experimental setup, everything presented in this paper can also be used in OBS-based methods. We leave that for future work.

## 3 Revisiting Loss Modelling for Unstructured Pruning

In this work, we investigate linear and diagonal quadratic models of the loss function and their performance when used for pruning neural networks. In our empirical study, we aim at answering the following questions:

1. How do criteria based on weight magnitude, or linear or quadratic models compare at preserving training loss (i.e. at solving the minimization problem in Equation 1)?
2. How does the locality assumption behind criteria based on linear and quadratic models affect their performances?
3. Do pruning criteria that are better at preserving the loss lead to better fine-tuned networks?

We now describe the linear and quadratic models we use, as well as the strategies to enforce locality of the pruning steps.

### 3.1 Linear and Quadratic Models

In current training strategies, regularization techniques such as early stopping or dropout (Srivastava et al., 2014) are often used to counteract overfitting. In these setups, there is no reason to assume that the training has converged, implying that the linear term in the Taylor expansion should not be neglected. Thus, one can build a pruning criterion similar to OBD that includes the gradient term in the quadratic model from Equation 2, leading to the following saliencies:[2]

$$\Delta\mathcal{L}^{QM}(\boldsymbol{\theta}, \Delta\boldsymbol{\theta}_k) \approx \left| \frac{\partial\mathcal{L}(\boldsymbol{\theta})}{\partial\boldsymbol{\theta}_k}^\top \Delta\boldsymbol{\theta}_k + \frac{1}{2}\mathbf{G}_{kk}(\boldsymbol{\theta})\Delta\boldsymbol{\theta}_k^2 \right| \Rightarrow s_k^{\text{QM}} = \left| -\frac{\partial\mathcal{L}(\boldsymbol{\theta})}{\partial\boldsymbol{\theta}_k}\boldsymbol{\theta}_k + \frac{1}{2}\mathbf{G}_{kk}(\boldsymbol{\theta})\boldsymbol{\theta}_k^2 \right| \quad (5)$$

Recall the constraint $\Delta\boldsymbol{\theta}_k \in \{-\boldsymbol{\theta}_k, 0\}$, hence the saliencies. This criterion generalizes OBD for networks that are not at convergence, and provides similar saliencies for networks that have converged.

To avoid the computational cost associated with computing second order information, which is prohibitive for large scale neural networks, one can use a simpler linear model (LM) instead of a quadratic one to approximate $\Delta\mathcal{L}(\boldsymbol{\theta}, \Delta\boldsymbol{\theta})$, leading to the following approximation and saliencies:

$$\Delta\mathcal{L}^{LM}(\boldsymbol{\theta}, \Delta\boldsymbol{\theta}) = \left| \frac{\partial\mathcal{L}(\boldsymbol{\theta})}{\partial\boldsymbol{\theta}}^\top \Delta\boldsymbol{\theta} \right| \quad \Rightarrow \quad s_k^{\text{LM}} = \left| \frac{\partial\mathcal{L}(\boldsymbol{\theta})}{\partial\boldsymbol{\theta}_k}\boldsymbol{\theta}_k \right| \quad (6)$$

The saliencies of the linear model are very related to the criterion used in Single-shot Network Pruning (Lee et al., 2019b), as demonstrated by Wang et al. (2020).

### 3.2 Enforcing Locality

One important point to keep in mind is that linear and quadratic models (whether diagonal or not) are *local* approximations, and are generally only faithful in a small neighbourhood of the current parameters. Explicitly showing the terms that are neglected, we have:

$$\Delta\mathcal{L}(\boldsymbol{\theta}, \Delta\boldsymbol{\theta}) = \Delta\mathcal{L}^{LM}(\boldsymbol{\theta}, \Delta\boldsymbol{\theta}) + \mathcal{O}(\|\Delta\boldsymbol{\theta}\|_2^2) = \Delta\mathcal{L}^{QM}(\boldsymbol{\theta}, \Delta\boldsymbol{\theta}) + \mathcal{O}(\|\Delta\boldsymbol{\theta}\|_2^3) \quad (7)$$

---

[2]Note that this idea has been explored for OBS (Singh & Alistarh, 2020), as well as for structured pruning (Molchanov et al., 2019)

So when approximating $\Delta\mathcal{L}$ with $\Delta\mathcal{L}^{LM}$ we neglect the terms in $\mathcal{O}(\|\Delta\boldsymbol{\theta}\|_2^2)$, and when approximating $\Delta\mathcal{L}$ with $\Delta\mathcal{L}^{QM}$ we neglect the terms in $\mathcal{O}(\|\Delta\boldsymbol{\theta}\|_2^3)$. Both approximations are thus only valid in a small neighbourhood of $\boldsymbol{\theta}$, and are extremely likely to be wrong when $\|\Delta\boldsymbol{\theta}\|_2$ is large. We list here different tricks to prevent this from happening.

**Performing the Pruning in several Stages**  $\|\Delta\boldsymbol{\theta}\|_2$ can be large when a large portion of the parameters is pruned at once. An easy fix typically used to mitigate this issue is to perform the pruning in several stages, re-estimating the model at each stage. The number of stages, which we denote by $\pi$, is typically overlooked (e.g. both Zeng & Urtasun (2019) and Wang et al. (2019) use only 6 stages of pruning). Our experiments, in upcoming Section 5, show that it has a drastic impact on the performances. Note that, without fine-tuning phases between the different pruning stages, this strategy violates the *convergence* assumption behind OBD and OBS, since after the first stage of pruning the network is no more at convergence.

The sparsity at each stage can be increased following either a linear schedule, where each step prunes the same number of parameter, or an exponential schedule, where the number of parameters pruned at each stage gets smaller and smaller. The later is typically used in the literature (Zeng & Urtasun, 2019; Wang et al., 2019; Frankle & Carbin, 2018; Renda et al., 2020). We compare them in Section 5.

**Constraining the Step Size**  As is often done when using quadratic models (e.g. Nocedal & Wright (2006)), one can penalize the model when it decides to take steps that are too large, in order to stay in a region where we can trust the model. This can be done by simply adding the norm penalty $\frac{\lambda}{2}\|\boldsymbol{\theta}_k\|_2^2$ to the saliencies computed by any criterion (Equations 3, 5 or 6), where $\lambda$ is a hyper-parameter that controls the strength of the constraint: a small value of $\lambda$ leaves the saliencies unchanged, and a large value of $\lambda$ transforms the pruning criterion into MP (Equation 4).

**Other Considerations**  $\|\Delta\boldsymbol{\theta}\|_2$ can be large if $\boldsymbol{\theta}$ is large itself. This is dependent on the training procedure of the network, but can be easily mitigated by constraining the norm of the weights, which can be done using $L_2$ regularisation or weight decay. Since nowadays weight decay is almost systematically used by default when training networks (e.g. He et al. (2016b); Xie et al. (2017); Devlin et al. (2018)), we do not investigate this further.

## 4 METHODOLOGY

We follow the main recommendations from Blalock et al. (2020). For fair comparison between criteria, all experiments are from our own PyTorch (Paszke et al., 2017) re-implementation, and ran on V100 GPUs. We use 5 different random seeds, and both mean and standard deviations are reported. We experiment with a MLP on MNIST, and with both VGG11 (Simonyan & Zisserman, 2014) and a pre-activation residual network 18 (He et al., 2016b) on CIFAR10 (Krizhevsky et al., 2009), to have variability in architectures, while using networks with good performance to number of parameters ratio. We further validate our findings on ImageNet (Deng et al., 2009) using a residual network 50 (He et al., 2016a). Although MNIST is not considered a good benchmark for pruning (Blalock et al., 2020), it can still be used to compare the ability of different criteria to solve the minimization problem in Equation 1. See Appendix B for details about splits, data augmentation and hyper-parameters.

**Pruning Framework**  Algorithm 1 presents the pruning framework used in this work: we first train the network, then perform several stages of pruning, and finally perform a single phase of fine-tuning, using the same hyper-parameters as for the original training. *Global* pruning is used for all the criteria. Note that because of their convergence assumption, OBD and OBS advocate for fine-tuning after each stage of pruning. Since LM and QM are not based on this assumption, they should perform well in this proposed framework. While the fine tuning-phase would require hyper-parameters optimisation, Renda et al. (2020) showed that using the same ones as for the original training usually leads to good results. The hyper-parameters used in our experiments are provided in Appendix B.

**Performance Metrics**  The performances of the pruning criteria are measured using two metrics: First, we use $\Delta\mathcal{L}(\boldsymbol{\theta}, \Delta\boldsymbol{\theta}) = |\mathcal{L}(\boldsymbol{\theta} + \Delta\boldsymbol{\theta}) - \mathcal{L}(\boldsymbol{\theta})|$, which is the quantity that the pruning criteria

---

**Algorithm 1** Pruning Framework

---

**Require:** Network $f_{\boldsymbol{\theta}}$ with $\boldsymbol{\theta} \in \mathbb{R}^D$, dataset $\mathcal{D}$, number of pruning iterations $\pi$, and sparsity $\kappa$.
 1: $f_{\boldsymbol{\theta}} \leftarrow \text{Training}(f_{\boldsymbol{\theta}}, \mathcal{D})$
 2: $\kappa_0 \leftarrow 0$
 3: $\mathbf{m} \leftarrow \mathbf{1}^D$
 4: **for** $i = 1$ to $\pi$ **do**
 5:      $\kappa_i \leftarrow \kappa_{i-1} + \frac{(\kappa - \kappa_0)}{\pi}$ **or** $\kappa_i \leftarrow \kappa_{i-1} + (\kappa - \kappa_0)^{i/\pi}$      ▷ Compute sparsity for iteration $i$
 6:      $\boldsymbol{s} \leftarrow \text{Saliencies}(f_{\boldsymbol{\theta} \odot \mathbf{m}}, \mathcal{D})$      ▷ Compute saliencies (Equation 3, 4, 5 or 6).
 7:      $\mathbf{m}[\text{argsort}(\boldsymbol{s})[: \kappa_i D]] \leftarrow 0$      ▷ Mask the parameters with smallest saliencies.
 8: $f_{\boldsymbol{\theta} \odot \mathbf{m}} \leftarrow \text{Training}(f_{\boldsymbol{\theta} \odot \mathbf{m}}, \mathcal{D})$      ▷ Optional fine-tuning
 9: **return** $f_{\boldsymbol{\theta} \odot \mathbf{m}}, \mathbf{m}$

---

are designed to minimize (recall Equation 1). Second, we use the validation error gap before/after fine-tuning, which is the metric we ultimately care about when designing pruning methods.

## 5   PERFORMANCES BEFORE FINE-TUNING

We evaluate the impact of enforcing locality in the LM, QM and OBS criteria. For each criterion, Figure 1 reports $\Delta \mathcal{L}(\boldsymbol{\theta}, \Delta \boldsymbol{\theta})$ as a function of $\lambda$, for different number of pruning stages $\pi$, using the exponential pruning schedule, and Figure 5 in Appendix show the same results for the linear pruning schedule. A typical usage of these criteria would be with a regularisation strength $\lambda = 0$ and a number of pruning stages $\pi \approx 1$. MP, the baseline, which is invariant to both $\lambda$ and $\pi$, is also reported in dashed black. For reference, the networks reached a validation error rate before pruning of $1.47 \pm 0.04$ % for the MLP, $10.16 \pm 0.29$ % for VGG11 and $4.87 \pm 0.04$ % for the PreActResNet18.

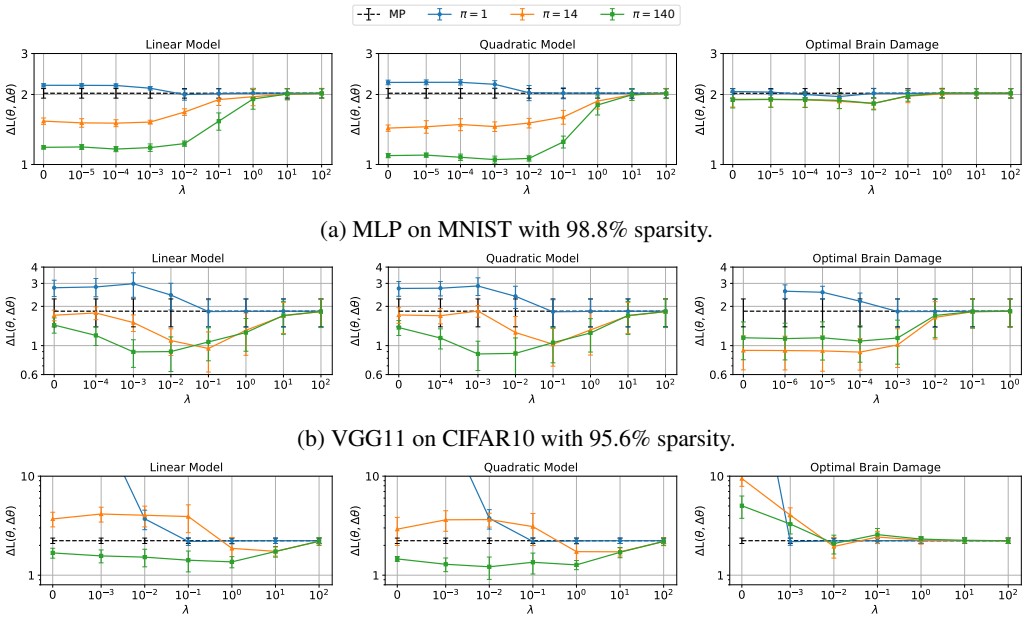

(a) MLP on MNIST with 98.8% sparsity.

(b) VGG11 on CIFAR10 with 95.6% sparsity.

(c) PreActResNet18 on CIFAR10 with 95.6% sparsity.

Figure 1: $\Delta \mathcal{L}(\boldsymbol{\theta}, \Delta \boldsymbol{\theta})$ for different number of pruning stages $\pi$, as a function of $\lambda$, the step size constraint strength, using either (left) LM, (middle) QM or (right) OBD criteria. MP, which is invariant to $\lambda$ and to the number of pruning stages, is displayed in dashed black. The curves are the mean and the error bars the standard deviation over 5 random seeds. OBD with $\pi = 1$ and $\lambda = 0$ diverged for all of the 5 seeds. Increasing the number of pruning stages drastically reduces $\Delta \mathcal{L}(\boldsymbol{\theta}, \Delta \boldsymbol{\theta})$. A $\lambda > 0$ can also help improving performances. Figure 6 in Appendix contains the same plots, but displaying the validation gap before/after pruning.

## 5.1 Impact of the Assumptions behind the Different Criteria

**Locality Assumption** Figure 1 shows that increasing the number of pruning stages can drastically reduce $\Delta\mathcal{L}(\boldsymbol{\theta}, \Delta\boldsymbol{\theta})$ when using LM, QM and OBS criteria. It demonstrates the importance of applying local steps when pruning. Constraining the steps size through $\frac{\lambda}{2}\|\boldsymbol{\theta}_k\|_2^2$ can also reduce $\Delta\mathcal{L}(\boldsymbol{\theta}, \Delta\boldsymbol{\theta})$, on CIFAR10 in particular. The trend, however, is less pronounced on MNIST. We hypothesize that it is due to the pruning step size: the MLP contains 260k parameters, vs 9.7M for VGG11, so the number of parameters pruned at each stage in VGG11 is still large, even with $\pi = 140$. This translates to a bigger $\|\Delta\boldsymbol{\theta}\|_2$ that needs to be controlled by the regularisation constraint.

**Convergence Assumption** When performing the pruning in several stages, we also observe that LM and QM can reach better performances than OBD. Without retraining phases between pruning stages, we violate the convergence assumption of OBD. This is however not the case for LM and QM, since they are not based on this assumption. Note that OBD still works reasonably well on VGG11. This could be be related to the depth of VGG11: VGG11 is deeper than the MLP, but not equipped with residual connections like the PreActResNet18.

## 5.2 Loss-preservation Capabilities of the Different Criteria

Table 1 contains the best $\Delta\mathcal{L}(\boldsymbol{\theta}, \Delta\boldsymbol{\theta})$ for each of the networks and pruning criteria. Our main observation is that the criteria that model the loss (LM and QM in particular) are better at preserving the loss than MP. Similarly to Table 1, Table 3 in Appendix contains the best validation error gap before/after pruning, where we can observe similar tendencies.

Table 1: Summary of the best $\Delta\mathcal{L}(\boldsymbol{\theta}, \Delta\boldsymbol{\theta})$ across values of $\lambda$ for different networks and pruning criteria, with $\pi = 140$. QM achieves better loss-preservation than other criteria. OBD performs worse than QM, since we violate its convergence assumption when pruning in several stages.

| Network | $\Delta\mathcal{L}(\boldsymbol{\theta}, \Delta\boldsymbol{\theta})$ | | | |
| --- | --- | --- | --- | --- |
| | **MP** | **OBD** | **LM** | **QM** |
| MLP on MNIST | $2.02 \pm 0.10$ | $1.83 \pm 0.11$ | $1.17 \pm 0.03$ | $\mathbf{1.05 \pm 0.04}$ |
| VGG11 on CIFAR10 | $1.84 \pm 0.44$ | $\mathbf{0.89 \pm 0.24}$ | $\mathbf{0.90 \pm 0.21}$ | $\mathbf{0.86 \pm 0.22}$ |
| PreActResNet18 on CIFAR10 | $2.23 \pm 0.14$ | $1.95 \pm 0.46$ | $1.36 \pm 0.18$ | $\mathbf{1.22 \pm 0.31}$ |

## 5.3 Linear vs Exponential Pruning Schedule

Figure 2 compares the impact of $\|\Delta\boldsymbol{\theta}\|_2$ and reports the training error gap when pruning VGG11 on CIFAR10 in several stages, using either the linear or the exponential pruning schedule. We also compare against one-shot pruning, as reference. The exponential schedule allows to maintain a more constant $\|\Delta\boldsymbol{\theta}\|_2$ throughout the pruning procedure, which limits the maximum size of $\|\Delta\boldsymbol{\theta}\|_2$, and thus better satisfies the locality assumption.

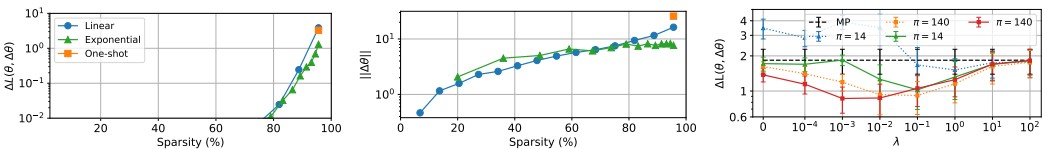

Figure 2: Linear vs exponential schedule using QM on VGG11. Left: $\Delta\mathcal{L}(\boldsymbol{\theta}, \Delta\boldsymbol{\theta})$ vs sparsity, zoomed on the end. Markers denote the 14 pruning stages. Middle: $\|\Delta\boldsymbol{\theta}\|_2$ at each stage. Right: Same as Figure 1, but comparing exponential (solid) and linear (dotted) schedules at 95.6% sparsity, with $\pi \in \{14, 140\}$. We get smaller $\|\Delta\boldsymbol{\theta}\|_2$ per pruning stage when using exponential instead of linear schedule, resulting in a smaller $\Delta\mathcal{L}(\boldsymbol{\theta}, \Delta\boldsymbol{\theta})$. It is advantageous to use that schedule when the pruning budged is limited, i.e. when $\pi$ is small. This advantage vanishes for larger values of $\pi$.

## 6    PERFORMANCES AFTER FINE-TUNING

We now fine-tune the pruned networks using the same hyper-parameters and number of epochs than for the original training. Table 2 shows the validation error gap between the non-pruned networks and the pruned networks after fine-tuning, for all considered criteria. LM performs better than MP on both the MLP and VGG11 (0.5% difference), but all criteria perform similarly on the PreActResNet18. These results are consistent with the observations of Blalock et al. (2020). As reference, global random pruning resulted in validation error rate of $47.18 \pm 6.8$ % for the MLP, and resulted in non-retrainable networks on CIFAR10 (with $90$ % error rate).

Table 2: Best validation error gap of the fine-tuned networks (lower is better), for different pruning criteria, across values of $\lambda$ and $\pi$. LM is better than MP on the MLP and VGG11. All the methods reach similar levels of performance on the PreActResNet18.

| Network | Gap of Validation Error (%) | | | |
|---|---|---|---|---|
| | MP | OBD | LM | QM |
| MLP on MNIST | $2.4 \pm 0.3$ | $2.0 \pm 0.1$ | $\mathbf{1.9 \pm 0.3}$ | $\mathbf{1.9 \pm 0.2}$ |
| VGG11 on CIFAR10 | $0.2 \pm 0.2$ | $-0.1 \pm 0.2$ | $\mathbf{-0.3 \pm 0.1}$ | $-0.1 \pm 0.1$ |
| PreActResNet18 on CIFAR10 | $0.2 \pm 0.2$ | $0.2 \pm 0.2$ | $\mathbf{0.1 \pm 0.1}$ | $0.2 \pm 0.2$ |

### 6.1    CORRELATION BETWEEN LOSS-PRESERVATION AND PERFORMANCES AFTER FINE-TUNING

An important observation is that the hyper-parameters $\lambda$ and $\pi$ that give the best performing criteria in terms of $\Delta\mathcal{L}(\boldsymbol{\theta}, \Delta\boldsymbol{\theta})$ in Table 1 are not the same as the ones that give the best performing criteria after fine-tuning in Table 2. We display in Figure 3 scatter plots of all the experiments we ran, to observe how well loss-preservation correlates with performance after fine-tuning.

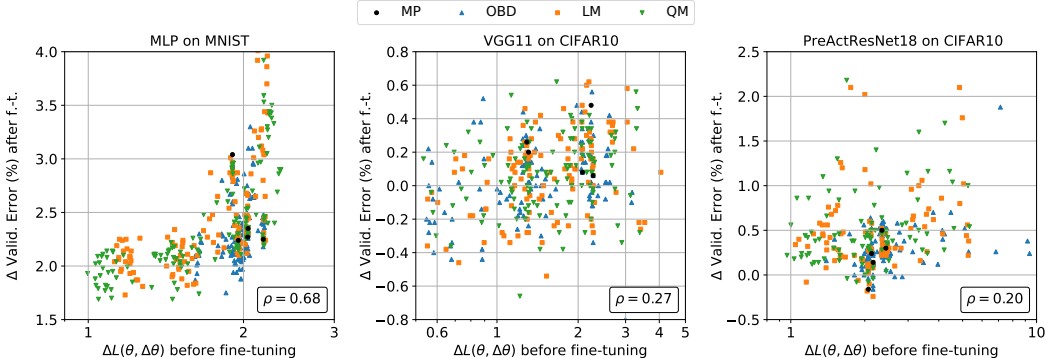

Figure 3: Gap of validation error after fine-tuning as a function of $\Delta\mathcal{L}(\boldsymbol{\theta}, \Delta\boldsymbol{\theta})$. Each point is one experiment, i.e. one one random seed, one $\pi$ and one $\lambda$. $\rho$ is the Spearman's rank correlation coefficient computed on all the data points. Except for the MLP on MNIST, there is only a weak correlation between $\Delta\mathcal{L}(\boldsymbol{\theta}, \Delta\boldsymbol{\theta})$ and the gap of validation after fine-tuning. Thus, the performance after pruning cannot be explained solely by the loss-preserving abilities of the pruning criteria.

Quite surprisingly, although we are able to obtain networks with smaller $\Delta\mathcal{L}(\boldsymbol{\theta}, \Delta\boldsymbol{\theta})$, and thus better performing networks right after pruning, the performances after fine-tuning do not correlate significantly with the gap. Except for the MLP on MNIST, whose Spearman's rank correlation coefficient is $\rho = 0.67$, there is only a weak correlation between $\Delta\mathcal{L}(\boldsymbol{\theta}, \Delta\boldsymbol{\theta})$ and the validation error gap after fine-tuning ($\rho = 0.27$ for VGG11 and $\rho = 0.20$ for PreActResNet18). Figure 10 in Appendix contains the same scatter plots, but showing $\mathcal{L}(\boldsymbol{\theta} \odot \mathbf{m})$ after fine-tuning instead of the validation error gap, and similar trends can be observed. Figure 11, also in Appendix, shows similar scatter plots, but for different sparsity levels on VGG11. Finally, Figure 9 in Appendix contains the same scatter plots but displaying the validation error gap before fine-tuning versus the validation error gap before fine-tuning.

To verify that these observations are not due to a specific choice of fine-tuning hyper-parameters, we perform a hyper-parameter grid search and report similar results in Appendix C.2. Also, we show in Figure 15 in Appendix C.3 the fine-tuning curves of networks pruned using MP and our best QM criteria. We observe that, except for MNIST, the difference in training loss right after pruning disappears after only one epoch of fine-tuning, erasing the advantage of QM over MP.

## 6.2 DISCUSSION

These results highlight an important issue: minimizing $\Delta \mathcal{L}(\boldsymbol{\theta}, \Delta \boldsymbol{\theta})$, no matter what model is used, might be used to design better pruning criteria, but it does not necessarily transfer to a better pruning method when fine-tuning is involved. The performance after fine-tuning cannot be explained solely by the local loss-preserving abilities of the criteria, and other mechanisms might be at play. Thus, the effect of fine-tuning should also be taken into account when designing pruning criteria.

For instance, Lee et al. (2019a) and Wang et al. (2020) proposed different heuristics to take into account gradient propagation in the context of *foresight* pruning, i.e. pruning untrained networks right after initialisation. Wang et al. (2020) argues that minimizing $\Delta \mathcal{L}(\boldsymbol{\theta}, \Delta \boldsymbol{\theta})$ in that context makes little sense, since the network is producing random predictions. In Appendix C.4 we compare our results to two pruning methods based on preserving the gradient flow, GraSP (Wang et al., 2020) and SynFlow (Tanaka et al., 2020), and show that they comply with our observations above.

Finally, several recent articles are looking further into the impact of various pruning criteria on subsequent training or fine-tuning (Lubana & Dick, 2020; Evci et al., 2020; Frankle et al., 2020).

## 7 SCALING UP TO IMAGENET

To investigate whether our observations also hold on larger datasets, we perform similar experiments with LM, QM and OBD on the ResNet50 on ImageNet. Before pruning, the network reached 76.41% validation accuracy. Figure 4 presents results at 70 % sparsity, in a similar fashion as Figure 1 and Figure 3. We observe a similar trend: The best loss-preserving models are not necessarily the best models after fine-tuning. See Appendix B for the detailed experimental setting, and see Figure 17 for results at 90 % sparsity.

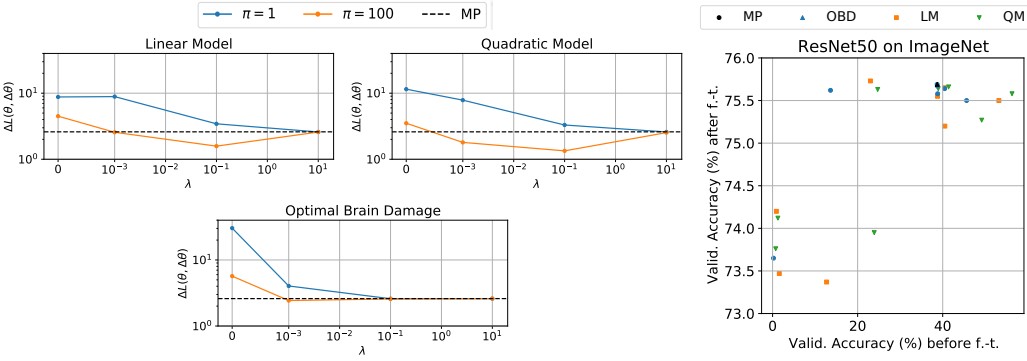

Figure 4: Same as Figure 1 and Figure 3, for the ResNet50 on ImageNet, with a sparsity of 70 %. Increasing the number of pruning stages and constraining the step size reduce $\Delta \mathcal{L}(\boldsymbol{\theta}, \Delta \boldsymbol{\theta})$. However, the best-loss preserving criteria, which maximize the validation accuracy right after pruning, do not produce better networks after fine-tuning. They perform similarly if their validation accuracy after pruning is $> 20\%$. Criteria that outperform MP right after pruning do not achieve better performance after fine-tuning.

## 8 CONCLUSION

In this paper, we revisited loss modelling for unstructured pruning. We showed that keeping the gradient term in the diagonal quadratic model allows to relax the convergence assumption behind

OBS and OBD. We also showed the importance of locality when using loss models for pruning: increasing the number of pruning stages and constraining the step size are two improvements that produce better loss-preserving pruning criteria and that should be added to the recommendation list of Blalock et al. (2020). Finally we observed that the loss right after pruning does not always correlate with the performances after fine-tuning, suggesting that a better loss before fine-tuning is not solely responsible for the performances after fine-tuning. Thus, future research should focus on ways to model the actual effect of subsequent fine-tuning when designing pruning criteria.

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

## APPENDIX

## A    GENERALIZED GAUSS-NEWTON

Having to compute $\mathbf{H}(\boldsymbol{\theta})$ is an obvious drawback of quadratic models, and thus a common first step is to approximate $\mathbf{H}(\boldsymbol{\theta})$ using the Generalized Gauss-Newton approximation (Schraudolph, 2002):

$$\mathbf{H}(\boldsymbol{\theta}) = \underbrace{\frac{1}{N} \sum_{i=1}^{N} \frac{\partial f_{\boldsymbol{\theta}}(x_i)}{\partial \boldsymbol{\theta}}^{\top} \nabla^2_{u=f_{\boldsymbol{\theta}}(x_i)} \ell(u, t_i) \frac{\partial f_{\boldsymbol{\theta}}(x_i)}{\partial \boldsymbol{\theta}}}_{\mathbf{G}(\boldsymbol{\theta}), \text{ the Generalized Gauss-Newton}} + \underbrace{\sum_{k}^{K} \frac{\partial \ell(u, t_i)}{\partial u_k}\bigg|_{u=f_{\boldsymbol{\theta}}(x_i)} \frac{\partial^2 f_{\boldsymbol{\theta}}(x_i)_k}{\partial \boldsymbol{\theta}^2}}_{\approx 0} \quad (8)$$

$$\approx \mathbf{G}(\boldsymbol{\theta}) \quad (9)$$

where K is the number of outputs of the network. $\mathbf{G}(\boldsymbol{\theta})$ has the advantage of being easier to compute and is also positive semi-definite by construction.

## B    DETAILS ON THE EXPERIMENTAL SETUP

### B.1    SETUP

**Datasets**    We use the MNIST dataset (LeCun et al., 1998), and hold-out 10000 examples randomly sampled from the training set for validation. We also use CIFAR10 (Krizhevsky et al., 2009), where the last 5000 examples of the training set are used for validation, and we apply standard data augmentation (random cropping and flipping, as in He et al. (2016b)) during training phases. For ImageNet (Deng et al., 2009), we follow the experimental setting of Goyal et al. (2017).

**Network Architectures**    On MNIST, we use a MLP of dimensions 784-300-100-10, with Tanh activation functions. On CIFAR10, we use both: a VGG11 (Simonyan & Zisserman, 2014), equipped with ReLUs (Nair & Hinton, 2010), but no Batch Normalisation (Ioffe & Szegedy, 2015); and the PreActResNet18, which is the 18-layer pre-activation variant of residual networks (He et al., 2016b). MLP leverages Glorot & Bengio (2010) as initialization while the the weights of VGG11 and PreActResNet18 are initialized following He et al. (2015), and the biases are initialized to 0. On ImageNet (Deng et al., 2009), we use a ResNet-50 (He et al., 2016a) with Batch Normalization, and follow the initialization strategy described in (Goyal et al., 2017).

### B.2    EXPERIMENTS

For the MNIST and CIFAR10 experiments, the network is first trained for a fixed number of epochs, using early stopping on the validation set to select the best performing network.The hyper-parameters used for training are selected via grid search (before even considering pruning). Then we prune a large fraction of the parameters. For OBD, LM and QM, we randomly select, at each iteration of pruning, 1000 examples (10 mini-batches) from the training set to compute the gradients and second

order terms of the models.[3] Finally, we retrain the network using exactly the same hyper-parameters as for the initial training.

For ImageNet, we uses the exact same hyper-parameters than Goyal et al. (2017).

**MLP on MNIST** We train the network for 400 epochs, using SGD with learning rate of 0.01, momentum factor of 0.9, l2 regularisation of 0.0005 and a mini-batch size of 100. We prune 98.85% of the parameters.

**VGG11 on CIFAR10** We train the network for 300 epoch, using SGD with a learning rate of 0.01, momentum factor of 0.9, a l2 regularisation of 0.0005 and a mini-batch size of 100. The learning rate is divided by 10 every 60 epochs. We prune 95.6% of the parameters.

**PreActResNet18 on CIFAR10** We train the network for 200 epochs, using SGD with a learning rate of 0.1, momentum factor of 0.9, a l2 regularisation of 0.0005 and a mini-batch size of 100. The learning rate is divided by 10 every 70 epochs. We prune 95.6% of the parameters.

**ResNet50 on ImageNet** For ImageNet, we train a ResNet50 using 8 V100 GPUs. The total mini-batch size is 256, and we train our baseline network for 90 epochs. The learning rate schedule is identical to Goyal et al. (2017): a linear warm-up in the first 5 epochs and decay by a factor of 10 at epochs 30, 60 and 80. We then prune 70% of the parameters. After pruning, we fine-tune the models for 90 epochs using a learning rate of $1e^{-3}$. For LM, QM, OBD, we investigates the following hyper-parameter values: $\pi \in \{1, 100\}$, $\lambda \in \{1e^{-3}, 1e^{-1}, 0, 10, \}$. 1600 examples are used to compute the first and second order terms of the linear and quadratic models.

## C SUPPLEMENTARY RESULTS

### C.1 PERFORMANCES BEFORE FINE-TUNING

**Validation error Table** Table 3 is the same as Table 1, but containing the best validation error gap before/after pruning instead of $\Delta\mathcal{L}(\boldsymbol{\theta}, \Delta\boldsymbol{\theta})$. We can observe a similar trend as in Table 1: LM and QM give better performances than MP, and OBD performs poorly, since the convergence assumption is not respected.

Table 3: Best validation error gap before/after pruning for different networks and pruning criteria.

| Network | Gap of Validation Error (%) | | | |
|---|---|---|---|---|
| | **MP** | **OBD** | **LM** | **QM** |
| MLP on MNIST | $72.09 \pm 3.72$ | $64.89 \pm 5.74$ | $\mathbf{16.35 \pm 0.77}$ | $\mathbf{15.22 \pm 0.62}$ |
| VGG11 on CIFAR10 | $56.19 \pm 17.9$ | $18.84 \pm 5.54$ | $\mathbf{5.89 \pm 1.52}$ | $5.92 \pm 2.14$ |
| PreActResNet18 on CIFAR10 | $74.13 \pm 4.59$ | $49.08 \pm 8.18$ | $26.79 \pm 8.61$ | $\mathbf{21.48 \pm 5.96}$ |

**Linear pruning schedule** Figure 5 contains the same experiments than Figure 5, but using the linear schedule instead of the exponential one. There is a drastic difference in performances: One need roughly 10x more stages of pruning with the linear schedule to reach the training gap of the exponential schedule.

### C.2 PERFORMANCES AFTER FINE-TUNING

**Validation error figures** Figures 8 and 7 contain the same experiments than Figure 5, but displaying the validation error gap, for linear and exponential schedules, respectively. For completeness, Figure 6 shows the validation error gap before fine-tuning.

---

[3]Using 1000 examples or the whole training set made no difference in our experiments. Using less examples started to degrade the performances, which concord with the observations of Lee et al. (2019b)

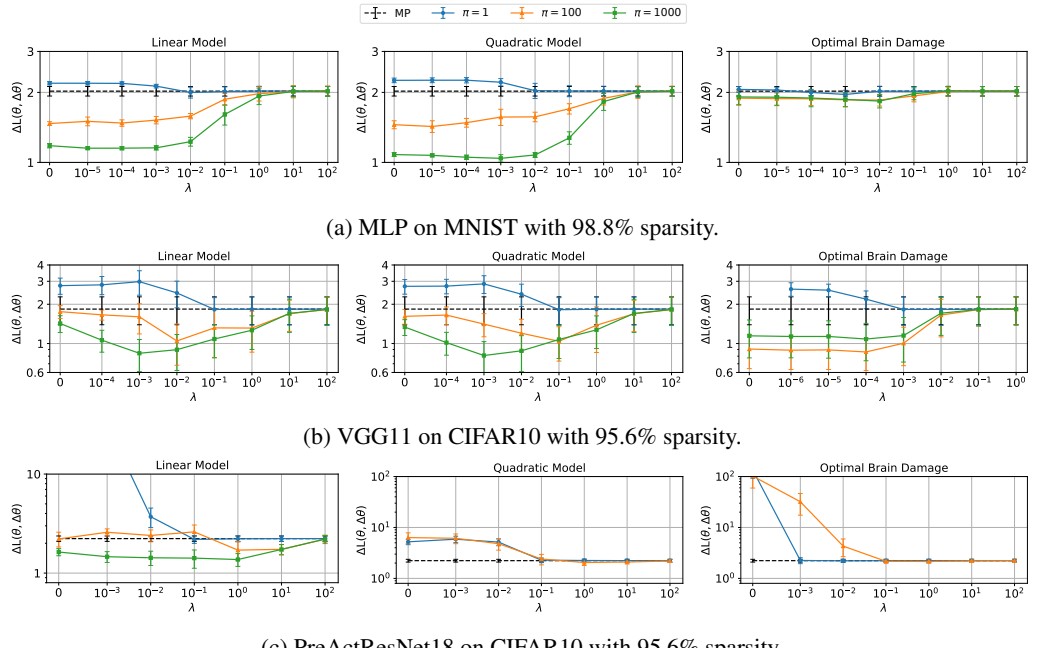

(a) MLP on MNIST with 98.8% sparsity.

(b) VGG11 on CIFAR10 with 95.6% sparsity.

(c) PreActResNet18 on CIFAR10 with 95.6% sparsity.

Figure 5: Same as Figure 1, but using equally spaced pruning steps. Note the difference in number of pruning iterations.

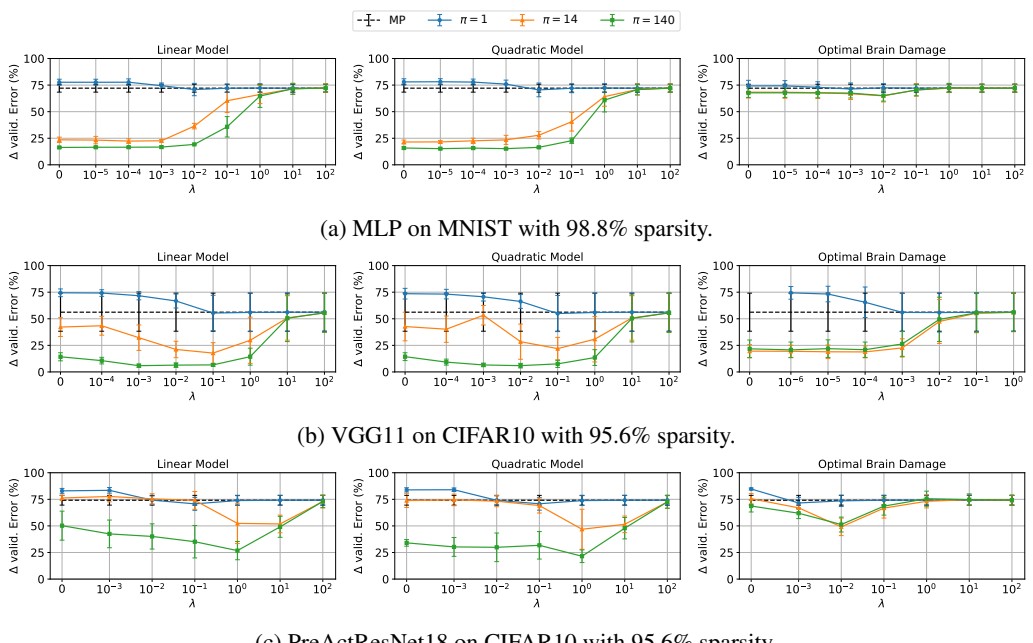

(a) MLP on MNIST with 98.8% sparsity.

(b) VGG11 on CIFAR10 with 95.6% sparsity.

(c) PreActResNet18 on CIFAR10 with 95.6% sparsity.

Figure 6: Same as Figure 1, but displaying the validation error gap before fine-tuning. With proper number of pruning stages and step size regularization, LM and QM can produce pruned networks that are drastically better than the ones pruned using MP.

**Validation gap before and after fine-tuning** Figure 9 is the same as Figure 3, but showing the validation error gap after fine-tuning as a function of the validation error gap before fine-tuning. As for Figure 3, we do not observe much correlation between the validation error before and after the fine-tuning.

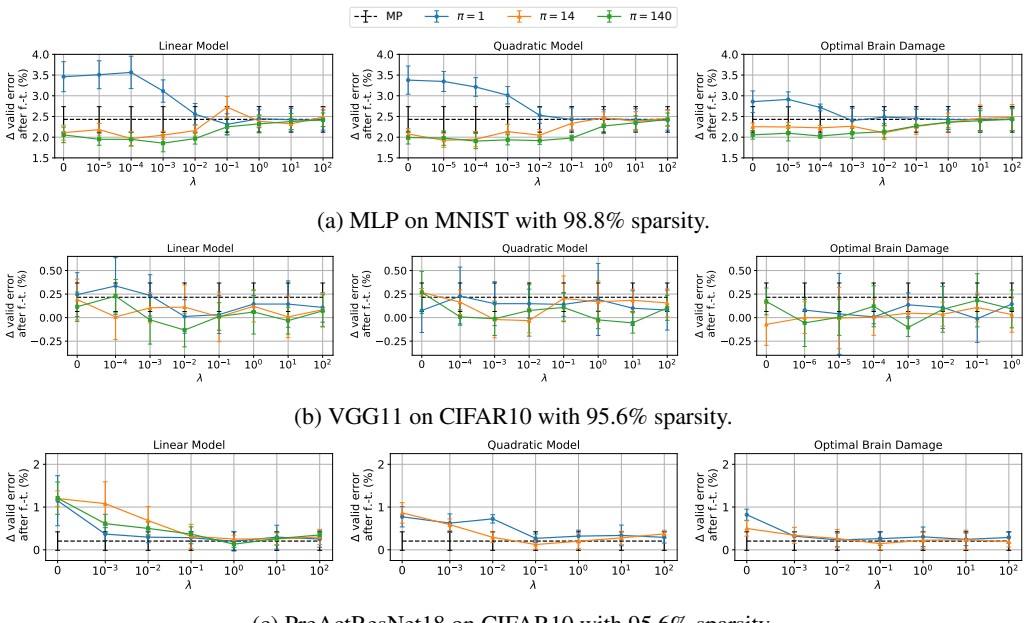

(a) MLP on MNIST with 98.8% sparsity.

(b) VGG11 on CIFAR10 with 95.6% sparsity.

(c) PreActResNet18 on CIFAR10 with 95.6% sparsity.

Figure 7: Same as Figure 1, but displaying the validation error gap after fine-tuning.

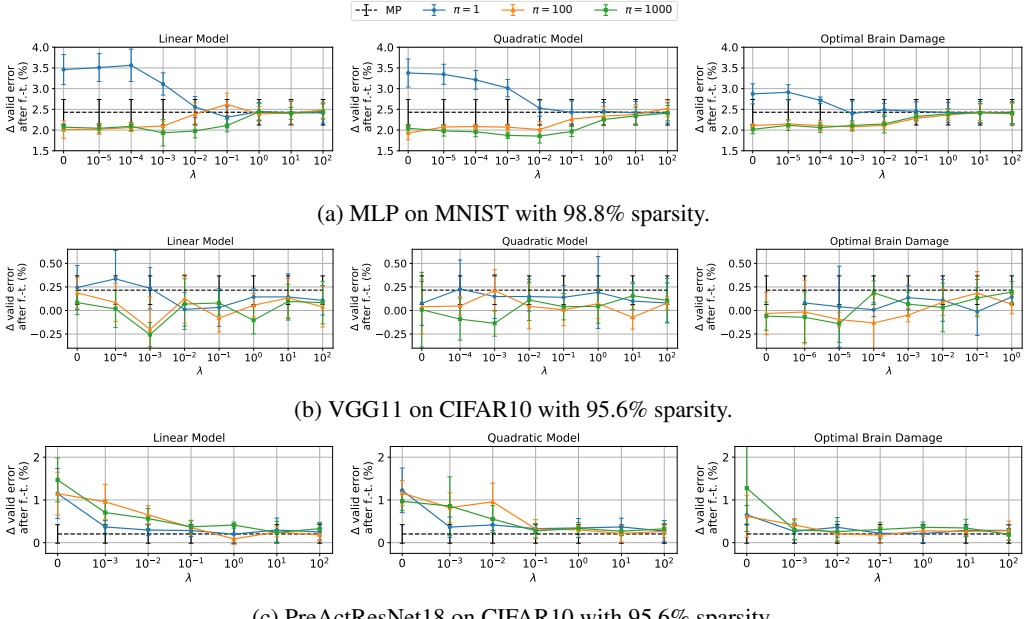

(a) MLP on MNIST with 98.8% sparsity.

(b) VGG11 on CIFAR10 with 95.6% sparsity.

(c) PreActResNet18 on CIFAR10 with 95.6% sparsity.

Figure 8: Same as Figure 7, but using equally spaced pruning steps. Note the difference in number of pruning stages.

**Training loss after fine-tuning** Figure 10 is the same as Figure 3 but showing $\mathcal{L}(\boldsymbol{\theta} \odot \mathbf{m})$ after fine-tuning as a function of $\Delta \mathcal{L}(\boldsymbol{\theta}, \Delta \boldsymbol{\theta})$. It has a similar trend as Figure 3: there is not much correlation between the loss before and after fine-tuning, except on MNIST.

**Different sparsity levels** Figure 11 shows the performances of different criteria on VGG11 on CIFAR10, for different sparsity levels. When the sparsity is low (89.3 %), the network has enough capacity to return to its original performances after fine-tuning. When the sparsity is too high (98.6 %),

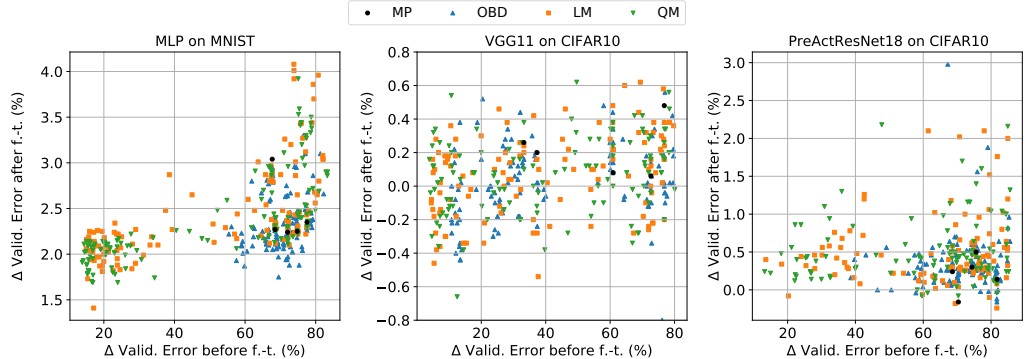

Figure 9: Same as Figure 3, but showing the validation error gap after fine-tuning as a function of the validation error gap before fine-tuning. Networks with drastically different performance before fine-tuning can still produce similar performances after fine-tuning.

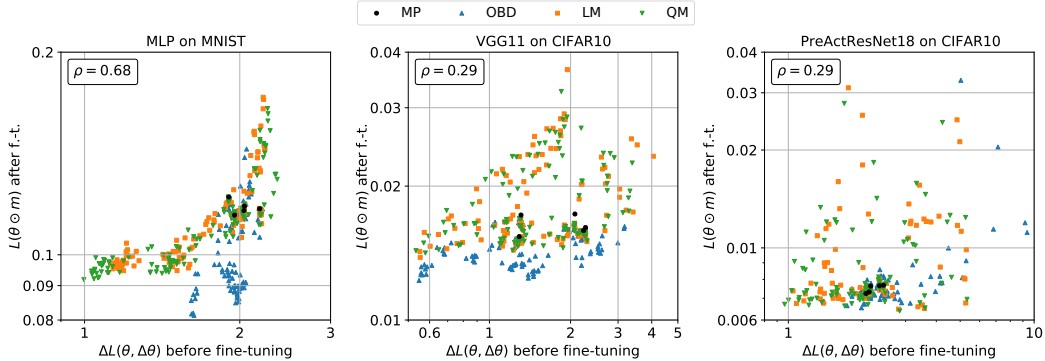

Figure 10: Same as Figure 3, but showing $\mathcal{L}(\boldsymbol{\theta} \odot \mathbf{m})$ after fine-tuning as a function of $\Delta\mathcal{L}(\boldsymbol{\theta}, \Delta\boldsymbol{\theta})$. Except for the MLP on MNIST, there is only a weak correlation between $\Delta\mathcal{L}(\boldsymbol{\theta}, \Delta\boldsymbol{\theta})$ and $\mathcal{L}(\boldsymbol{\theta} \odot \mathbf{m})$ after fine-tuning.

then all criteria produce networks with random predictions. There might be a sweet spot in between, but one would require more powerful model to verify this supposition.

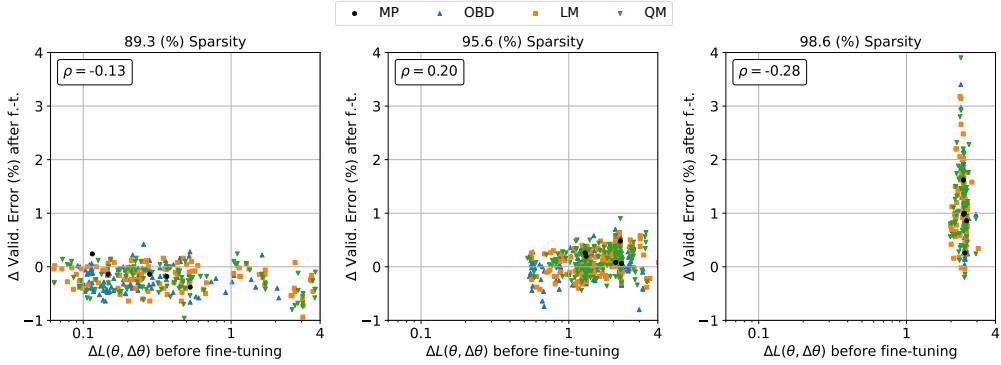

Figure 11: Same as Figure 3, but for different sparsity levels on the VGG11 on CIFAR10. When the sparsity is low, the network has enough capacity to return to its original performances after fine-tuning. When the sparsity is too high, then all criteria produce networks with random predictions.

**Hyper-parameters optimisation** Figure 13 shows the impact of hyper-parameter optimization for the fine-tuning phase. We performed a grid search with three different learning rate (0.1, 0.01, 0.03)

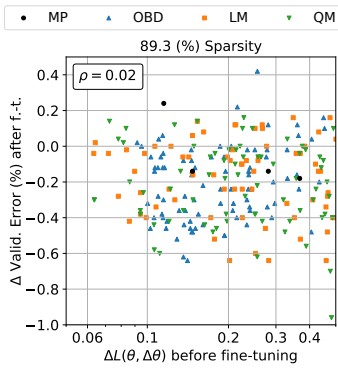

Figure 12: Same as Figure 11 (left), but zoomed on smaller values of $\Delta\mathcal{L}(\boldsymbol{\theta}, \Delta\boldsymbol{\theta})$.

and three different l2-regularisation (0, 5e-4, 5e-5). All 9 sets of hyper-parameters were tested on LM, QM and MP on 5 different random seeds. In this set of experiments, we used $\lambda \in \{0, 0.01, 0.1, 1\}$ and $\pi \in \{14, 140\}$. Optimizing hyper-parameters for fine-tuning can lead to better performance after fine-tuning, but does not increases the correlation between the performances after fine-tuning and $\Delta\mathcal{L}(\boldsymbol{\theta}, \Delta\boldsymbol{\theta})$. The lack of correlation can thus not be explained by bad fine-tuning hyper-parameters.

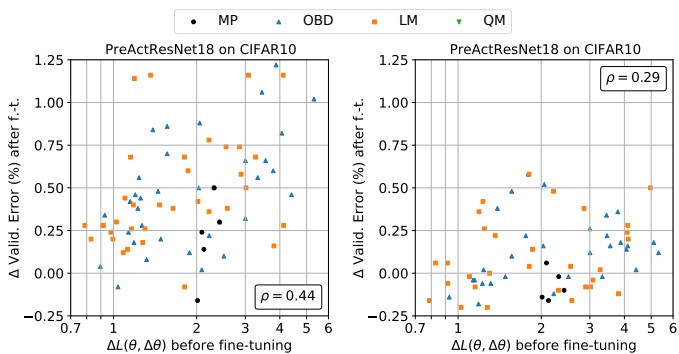

Figure 13: Left: Using the same hyper-parameters for fine-tuning as the ones of the original training. Right: Performing hyper-parameters optimisation for the fine-tuning. This figure shows that optimizing the hyper-parameters for fine-tuning can improve the performances of the network after pruning. However, it reduces the correlation between the performances after fine-tuning and $\Delta\mathcal{L}(\boldsymbol{\theta}, \Delta\boldsymbol{\theta})$. The lack of correlation can thus not be explained by poor fine-tuning hyper-parameters.

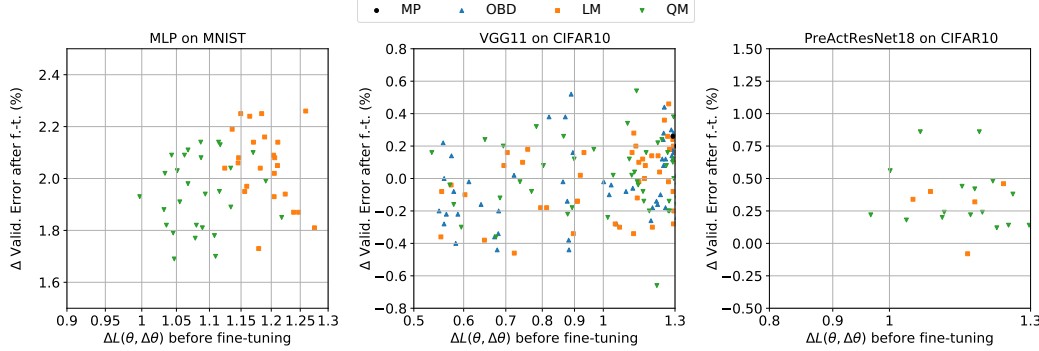

Figure 14: Same as Figure 3, but zooming on the best performing networks in terms of $\Delta\mathcal{L}(\boldsymbol{\theta}, \Delta\boldsymbol{\theta})$.

## C.3   FINE-TUNING CURVES

To investigate whether one of the networks is suffering from optimization issues during fine-tuning, we show in Figure 15 the fine-tuning curves of networks pruned using MP and our best QM criteria. We observe that, except for MNIST, the difference in training loss right after pruning disappears after only one epoch of fine-tuning, erasing the advantage of QM over MP.

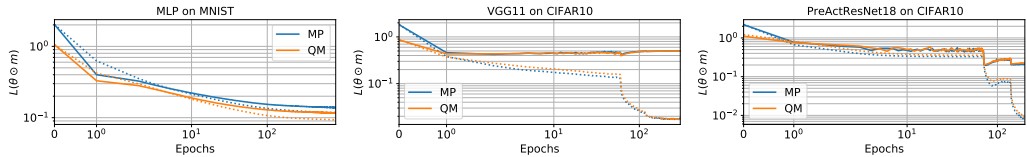

Figure 15: Fine-tuning losses (dotted is training, solid is validation) of networks pruned using MP and QM criteria. All the curves are the average over the 5 seeds. We do not show the standard deviation for clarity. Left: MLP, middle: VGG11 and right: PreActResNet18. Except for MNIST, the difference in loss right after pruning (i.e. at epoch 0) disappears after one epoch of fine-tuning.

## C.4   RESULTS USING GRASP AND SYNFLOW

We compare our results with two additional pruning methods that focus on preserving the flow of the gradient in the network instead of preserving the loss: GraSP (Wang et al., 2020), a *data-dependant* method, and SynFlow (Tanaka et al., 2020), a *data-agnostic* one. Both methods were design to be applied at initialisation, so we investigate here their use on trained networks. We use $\pi \in \{1, 100, 1000\}$, and added our proposed step size constraint $\lambda$ to the pruning criteria as well.

Figure 16 shows the scatter plot of the preservation of the loss vs the performance after fine-tuning. Similarly to what we observed before, there is no clear evidence that better preserving the loss lead to better performance after fine-tuning.

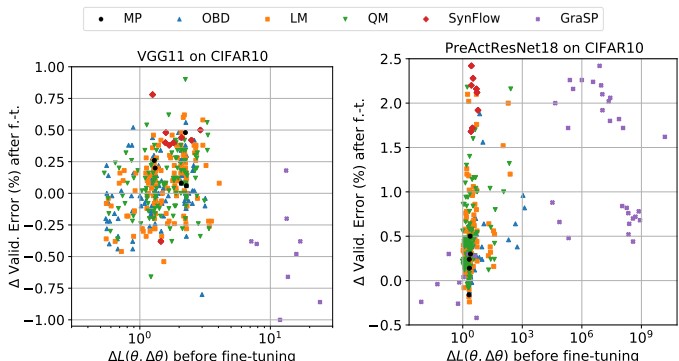

Figure 16: Same as Figure 3 showing GraSP and SynFlow on VGG11 (left) and the PreActResnet18 (right). This Figure shows that one can observe a large $\Delta\mathcal{L}(\boldsymbol{\theta}, \Delta\boldsymbol{\theta})$ and yet obtain very good performance after fine-tuning. This is especially true in the case of GraSP for VGG11. Furthermore, we can observe similar behaviour on PreActResNet18 where two different methods can lead to similar performance after fine-tuning while having completely different $\Delta\mathcal{L}(\boldsymbol{\theta}, \Delta\boldsymbol{\theta})$: GraSP with $\Delta\mathcal{L}(\boldsymbol{\theta}, \Delta\boldsymbol{\theta}) \approx 10^8$ has fine-tuning performance similar to MP with $\Delta\mathcal{L}(\boldsymbol{\theta}, \Delta\boldsymbol{\theta}) < 10^1$.

## C.5   RESULTS ON IMAGENET

Figure 17 is the same as Figure 4, but with 90 % sparsity. At that sparsity level, the validation accuracy right after pruning is close to random for all the pruning criteria. There is however quite a big variation in performances after fine-tuning: at equal performance before fine-tuning, some models achieve 70 % validation accuracy after fine-tuning, while others only reach 60 %.

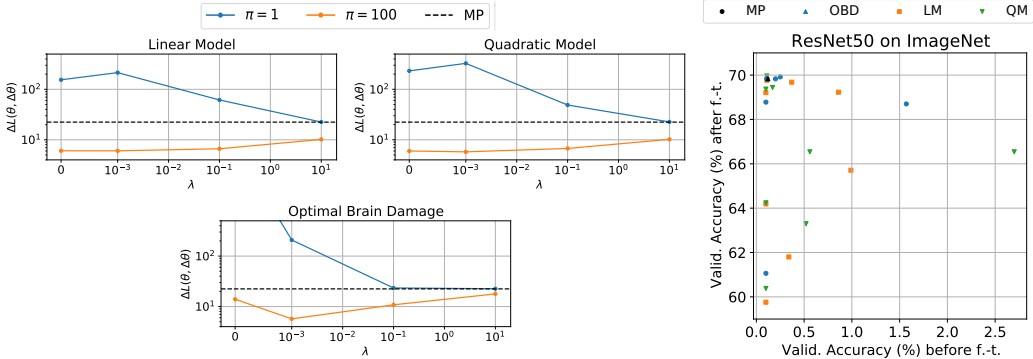

Figure 17: Same as Figure 4, but with 90 % sparsity. Increasing the number of pruning stages and constraining the step size reduce $\Delta\mathcal{L}(\boldsymbol{\theta}, \Delta\boldsymbol{\theta})$. However, the best-loss preserving criteria, which maximize the validation accuracy right after pruning, do not produce better networks after fine-tuning.

