# OpenReview forum: "Revisiting Loss Modelling for Unstructured Pruning"
_ICLR.cc/2021/Conference — Reject_

### Official Review · AnonReviewer2 · 2020-10-27
**Interesting paper with solid empirical evaluation**

**Rating:** 7
**Confidence:** 5

**Review:**

Summary:

The authors study the use of loss-modeling to maintain model quality when inducing unstructured sparsity in deep neural networks. They study a range of different approximations and modifications that can help improve the quality of the approximation (taking local steps, avoid large changes in weight magnitude, avoiding assumptions about convergence). The authors conduct a thorough empirical investigation that yields practical observations for the design of future pruning techniques.

Pros:

The paper is well written and well organized and the empirical investigations are well done. The observations made by the authors are interesting and practically useful for the development of future pruning techniques. Namely,
1. That including first order terms in loss approximations can relax the convergence assumption behind some existing loss modeling approaches to enable more flexible application of the pruning algorithm.
2. The quality of the local loss approximation can be improved by taking a series of smaller pruning steps.
3. That loss-preservation does not necessarily translate into accuracy preservation.

Cons:

It would be nice to see experiments in domains other than computer vision. For example, language modeling with RNNs or Transformers. Results at a wider range of sparsity levels for ImageNet would also have been useful, as it seems possible that these techniques could perform differently for high sparsity (>90%) than they do for moderate sparsity (e.g., the 70% sparsity reported in Figure 4).

Comments:

Another ICLR 2021 submission is highly relevant to your investigation: https://openreview.net/forum?id=rumv7QmLUue. Their theoretical/empirical results appear to corroborate your conclusions that loss preservation is not necessarily the best metric to optimize for when you care about accuracy preservation.

---

> ### Author Response · Authors · 2020-11-14
> **Response to Reviewer2**
>
> Thank you for your review and valuable feedback! We appreciate that you find our observations interesting and practically useful for the development of future pruning techniques. We address your comments in the following.
>
> ## Additional Experiments:
>
> Thank you for your suggestion. We added an experiment on ImageNet with 90% sparsity and included them in Appendix C5 and Figure 17. We can observe a similar trend, the performances after pruning do not correlate  with the performances after fine-tuning.
>
> ## Another ICLR 2021 submission:
>
> Thank you for this suggestion: it is indeed extremely relevant, and we added it to the paper.
>
> Thank you again for your review and let us know if you have further questions.

---

> > ### Comment · AnonReviewer2 · 2020-11-24
> > **Reviewer Response 1**
> >
> > Thank you to the authors for their response. The additional ImageNet experiments are useful. I'd still encourage the authors to explore domains other than computer vision. In particular, a Transformer model of an RNN would make the empirical results more robust.

---

### Official Review · AnonReviewer3 · 2020-10-29
**Appreciate the empirical study but cannot find substantial useful conclusions**

**Rating:** 5
**Confidence:** 4

**Review:**

The paper develops two modified version of Optimal Brain Damage (OBD) criteria, namely LM and QM (linear and quadratic model) to measure the importance/saliency of model weights. It then compares these three together with Magnitude Pruning (MP), and show that these among four criteria: 1. for the first three, using iterative pruning to enforce locality of the gradient calculation is important, 2. the best method for training loss before fine-tuning does not necessarily lead to best validation accuracy after fine-tuning.

The paper's empirical investigation is valuable and appreciated. It helps me understand OBD more throughly and the assumptions behind it. It is also useful to know that using iterative pruning can improve these gradient approximation-based methods because of locality.

1. My primary concern is the experiments does not seem to lead to a useful guidance for future practice. The paper does conclude for the first three using iterative pruning is useful, but these three criteria are rarely used nowadays and MP is the mainstream, and the simplest method. The paper also didn't conclude which of the four criteria is in general best and recommended. From table 2 it seems to be LM but the paper did not conclude this way. This is also possibly due to that the experiments are not run extensively on different datasets and architectures.

2. The paper compared the training loss before fine-tuning, and validation acc. after fine-tuning. But I think validation acc. before fine-tuning is also a quantity worth investigating.

3. More importantly, the paper shows the training loss (before fine-tuning) and valid acc. (after fine-tuning) are not necessarily correlated, but did not give explanation on why this could be the case through experiments, or give useful suggestions to achieve a good valid acc. after fine-tuning.

Overall I appreciate the empirical study but I suggest conducting the experiments on more datasets and architectures, and extract a useful conclusion to guide future practices.

+++++++++++++++++

I appreciate the clarified messages of the paper, and would like to see them emphasized more clearly in the next version of the paper. But due to the limited experimental scale on ImageNet (added in rebuttal, and in my understanding, it only verifies one of the multiple observations mentioned in the paper), I'm still leaning on rejection. I updated my score from 4 to 5.

---

> ### Author Response · Authors · 2020-11-14
> **Response to Reviewer3**
>
> Thank you for your review and valuable comments. We are glad our paper helped understanding the assumptions behind using loss models for pruning, as it is one key message we wanted to tell the community.
>
> ## Useful guidance for future practice:
>
> The objective of the paper was to better understand the role of the assumptions behind the loss models used to design pruning criteria. We hypothesized that the bad performances of LM and OBD were due to violation of the assumptions behind them, and that was the reason why MP was performing better and became the mainstream pruning criterion, as you rightfully mentioned. We show that by better respecting the assumptions we can indeed get better performances after pruning, but this does not necessarily correlate to better performances after a subsequent fine-tuning. So here is guidance that can be extracted from these results:
>
> 1. If you want to select a method to prune a model, then MP is a good first criterion to use (and it works well on large scale experiments, as shown in Section 7). Now, understanding why some criteria work better than others in different cases is still an active area of research (e.g. Frankle et al., (2020)).
> 2. If you want to use loss-based criteria, you should pay a special attention to the locality and convergence assumptions, as they have a drastic impact on the performances (as shown in Section 5). However, you need to keep in mind that it is not clear how these criteria perform with subsequent fine-tuning (as shown in Section 6).
> 3. Most importantly for further research in pruning, if you want to design a new pruning criterion, we suggest not to look solely at the impact that the pruning has on the loss, as it doesn’t correlate with the performances after fine-tuning (as shown in Section 6).
>
> ## Validation accuracy before fine-tuning:
>
> We primarily showed the training loss before fine-tuning, as it is the objective we are trying to minimize (Equation 1). To provide a complete picture, we added the plots showing the gap in validation accuracy before fine-tuning in the Appendix (Figure 6 and Figure 9). Thanks for the suggestion!
>
> ## Explanation of why the training loss (before fine-tuning) and valid accuracy (after fine-tuning) are not necessarily correlated, or give useful suggestions to achieve a good valid accuracy after fine-tuning.
>
> We would like to point out that the discovery that the training loss (before fine-tuning) and valid accuracy (after fine-tuning) is a useful observation. Our work shows that there is a trade-off between function preserving capability and re-trainability: The parameters that preserve best the function might not be the one that ensure the good re-trainability of the network. To the best of our knowledge, network re-trainability has not been thoroughly investigated in the context of pruning and should be taken into account when designing a new pruning criteria. Some very recent papers are starting to look into it (e.g. Frankle et al. (2020), Lubana & Dick (2020), Evci et al. (2020)), but it remains an open question.
>
>
> ## Conducting the experiments on more datasets and architectures, and extract a useful conclusion to guide future practices.
>
> One goal when designing our empirical study was to ensure the robustness of our observations. We therefore explore an extensive range of hyper-parameters for all the different pruning approaches. As a result, our empirical study already contains more than 2500 experiments (each point in each scatter plot being one experiment).
>
>
> Thank you again for taking the time to review our paper. We hope that our answers clarified the main message as well as the useful guidance we can deduce from our empirical studies. We do hope that you will reconsider your score given those clarification. Let us know if you have other questions or concerns.

---

### Official Review · AnonReviewer4 · 2020-10-30
**A paper that combines many different things from other works, and runs a few extra experiments, but in the end doesn't feel like it has enough body and depth for a full-conference paper.**

**Rating:** 3
**Confidence:** 4

**Review:**

Although the paper is covering an interesting topic, much of what's in the paper can be found in other works, and there's not a lot of novelty to the insights, nor a large breadth of experiments to justify it as a survey paper.
- The linear and quadratic loss functions are not new.
- The enforcing locality part is essentially why the pruning strength annealing schemes exist, this insight is not new and can be found in Zhu&Gupta, or in Bayesian settings like in the Molchanov paper, they suggest annealing from a Bayesian statistical perspective
- The fact that this leads to multiple stages of pruning to be a good idea, is also known in the literature this paper cites.
- The most meat of the paper is in the 'survey' part of it, investigating the results... but this section feels lacking since there are not a lot of experiments, and the insights of e.g. post-finetuning can be largely found in e.g. Blalock et al. I'm missing deeper insights/analysis here. What is the reason for this? How do we remedy this? What are the characteristics of networks that lead to this behavior?
- It would have been great to see a lot more insight/experiments on this topic. The authors throw up a lot of hypothesis and suggestions/ideas throughout the paper, but don't back them up. E.g. If the ||Theta|| term is better of to be constant throughout the pruning procedure... can we somehow make an annealing scheme that keeps ||Theta|| small and constant throughout the pruning process, and show that that works well? This can be proven/shown somehow.

I do think the paper is well written; and I encourage the authors to look further into this topic and come up with more novel insights/results and methods to improve pruning

Other things/questions/suggestions:
- In formulations (1), (2) and (5), (6). Why are the absolute brackets necessary? Especially for models that have not converged, why would you want to stay close to the original loss, as opposed to just decreasing the overall loss of the model?
- For most of the discussion in section 3.2, the authors talk about the norm of the delta theta squared being large. But this largeness is relative to the 0th and first order term which the authors glance over. Under 'other considerations' for example, if the weights theta are large, the gradients likely follow suit. Thus the absolute magnitude of the weights might not matter, as it's the relative size of this to the gradient terms that should be considered.
- Constraining the step size in section 3.2. Interestingly, if you take the assumptions that for each layer, the hessian of the loss w.r.t. the output is sqrt(lambda/2), and your input distributions to a layer are normalized, you end up with the weaker norm penalty. This is a crude approximation of the second order term, which would give this method a bit more of a theoretical foundation than just a regularization term.
- 5.1 Convergence assumption. I don't get this part, both the OBD method, and the linear and quadratic loss terms depend on the locality, so all will also depend on the amount of steps taken. For OBD, as long as you recalculate the Gauss-Newton matrix, I don't see why this method is different when not doing fine-tuning.
- 5.1 Convergence assumption. The result cited in appendix A is a very well-know result. How could this link explain the OBD performance on the VGG network? 'Could' is not strong enough to make it into a paper.

Small tidbits:
3. Do pruning criteria better at preserving the loss lead to better fine-tuned networks? <- this sentence doesn't flow nicely. I would add a 'that' so you have do pruning criteria that are better ...

---

> ### Author Response · Authors · 2020-11-14
> **Response to Reviewer4 (1/2)**
>
> Thank you for taking the time to review the paper and your valuable feedback! We believe that there is a misunderstanding regarding the core contributions and novelty of our paper. We hope to clarify this in the present answer.
>
>
> ## There's not a lot of novelty to the insights ...
>
> We respectfully disagree. To the best of our knowledge, we are the first work that demonstrates the lack of correlation between training loss before fine-tuning and valid accuracy after fine-tuning when pruning a neural network. Our work shows that there is a trade-off between function preserving capability and re-trainability: The parameters that preserve best the function might not be the one that ensure the good re-trainability of the network.  In addition, to the best of our knowledge, network re-trainability has not been thoroughly considered in the context of pruning a trained network. Our findings suggest that it should be taken into account when designing a pruning criteria.
>
> Our work also highlights that if you want to use loss-based criteria, you should pay a special attention to their underlying assumptions, as they have a drastic impact on the performances
>
> We discuss the relation with (Blalock et al. (2020), Zhu & Gupta (2017), Molchanov et al. (2019)) below and  we would happily discuss the relation with other related works if you provide us with some references.
>
> ## ... nor a large breadth of experiments to justify it
>
> One goal when designing our empirical study was to ensure the robustness of our observations. We therefore explore an extensive range of hyper-parameters for all the different pruning approaches. As a result, our empirical study already contains more than 2500 experiments.
>
> ## The linear and quadratic loss functions are not new.
>
> We agree that LM and QM were previously used in pruning. The novel elements we bring to the community are:
> 1. A systematic comparison of LM, QM, OBD and MP against each other on several benchmarks, and this both before and after fine-tuning.
> 2. An analysis of the assumption behind these methods.
> 3. An empirical evaluation of the impact of such assumptions on the pruning performances, again both before and after fine-tuning.
>
> ## Relation with Zhu & Gupta (2017) and Molchanov et al. (2019)
>
> Zhu & Gupta (2017) and Molchanov et al. (2019) propose pruning algorithms where the pruning is performed during the training of the network. Thus, their rationale for pruning iteratively is to avoid disturbing the training procedure too much, not because of a locality assumption. Even more, Zhu & Gupta (2017) uses MP as a pruning criterion, and since MP is not based on a Taylor approximation, there is no locality assumption in their setup. In contrast, our rationale for pruning iteratively is to respect the locality assumption behind the models used in LM, QM and OBD criteria.
>
> Also we do not prune during training. Rather, we analyse the impact of re-estimating the loss model at each stage of the pruning, without any fine-tuning in between. Our setup is thus different from Zhu & Gupta (2017) and Molchanov et al. (2019).
>
> ## Multiple stages of pruning is also known in the literature this paper cites.
>
> We are not claiming it is a new method, but we rather analyze its importance. As we state in section 3.2: “The number of stages, which we denote by π, is typically overlooked (e.g. both Zeng & Urtasun (2019) and Wang et al. (2019) use only 6 stages of pruning).” We showed in our empirical evaluation that way more stages should be used instead, to properly enforce the locality assumption.
>
> ## Relation to Blalock et Al. (2020)
>
> Blalock et al. (2020) highlights that different fine-tuning hyper-parameters can yield variability in the final results for MP. In contrast, our work demonstrates that there is a lack of correlation between training loss before fine-tuning and valid accuracy after fine-tuning when pruning a neural network, i.e. the parameters that preserve best the function might not be the one that ensure the good re-trainability of the network.
>
> The fact that there is very little correlation between the performances before and after fine-tuning is a novel observation that brings the insight that focusing only on loss-preserving to design pruning criteria might not be a good strategy.
>
> Additionally,  Blalock et al. (2020)  does not analyse the impact of both locality and convergence assumption on any loss-based pruning criteria.

---

> > ### Author Response · Authors · 2020-11-14
> > **Response to Reviewer4 (2/2)**
> >
> > We address here your other comments.
> >
> > ## Annealing scheme that keeps $||\theta||$ small and constant throughout the pruning process, and show that that works well?
> >
> > We show in Figure 2 (middle) that the exponential pruning schedule keeps the $||\Delta \theta||$ relatively constant, and that it translates to a better performing network after pruning (Figure 2, left). It actually provides clear justification to use this scheme in practice.
> >
> > ## “The authors throw up a lot of hypothesis and suggestions/ideas throughout the paper, but don't back them up.”
> >
> > Could you point us to these unbaked hypotheses? We would happily clarify/justify them.
> >
> > ## Absolute value in formulation
> >
> > We agree that removing the absolute value is an interesting direction to explore, as there shouldn’t be any reason to prevent the pruning to produce pruned networks with better loss.
> >
> > We explored pruning criteria that did not use the absolute value, but we found out that it was extremely hard to prune models without the absolute values. We hypothesise that the main issue with that is that removing parameters with the smallest (negative) sailencies would correspond to removing parameters that improve the most the loss, but these parameters could also be the ones for which the loss model is likely to be wrong. However, further research is needed to clearly understand this behavior.
> >
> > We also note that SNIP (Lee et al., 2019), Molchanov et al. (2017) and Molchanov et al. (2019) also use the absolute values in their formulation, although they do not apply the pruning on converged networks either.
> >
> > ## Constraining the step size
> >
> > The idea of using L2 penalty for the locality assumption is actually coming from Trust Region methods, so we argue that as such it is already theoretically grounded. We selected it for its simplicity and for its interesting link with MP (a very high lambda allows us to recover MP). We totally agree that more advanced methods such as the one you suggest, or backtracking line search (as R5 proposed) could indeed be used instead. We were interested in investigating the impact of the locality assumption rather than investigating the best method to actually enforce it.
> >
> > ## 5.1 Convergence assumption of OBD.
> >
> > To clarify, OBD relies on the convergence assumption (i.e. the method assumes the gradients are zero). If the network is no more at convergence, the gradients are not zero and thus OBD is likely to be wrong. So even if the Gauss-Newton matrix is recomputed, the assumption of convergence behind OBD is still violated. LM and QM on the other hand do not rely on this assumption, as they incorporate the gradient term in their formulations. We see QM as a generalisation of OBD for models that are not at convergence.
> >
> > ## The result cited in appendix A
> >
> > The fact that OBD performs still relatively well on VGG is puzzling, since we violate its convergence assumption when pruning iteratively, and thus expect poor performances of OBD (such as in the MLP on MNIST and in the PreActResNet18 on CIFAR10). The only case where OBD could still perform well is when the 2nd order term of the loss landscape dominates the 1st order term.
> >
> > We agree the explanation in appendix A could be confusing and is unnecessary to the message of our paper, so we removed it.
> >
> > ## Small tidbits
> >
> > Thanks, it indeed flows better. We changed it!
> >
> >
> > Again, thank you for your review and valuable feedback. We hope that our answers clarify the significance and novelty of our contributions. We hope that you will reconsider our score which we believe is severe with respect to our work. Let us know if you have any other questions or remaining concerns.

---

### Official Review · AnonReviewer5 · 2020-11-07
**Well-written paper with interesting results.**

**Rating:** 6
**Confidence:** 5

**Review:**

Summary:
- This paper conducted a detailed study on how does the loss modeling affects the final performance of the pruned model. The authors first provided a unified view of various pruning algorithms (e.g., Magnitude Pruning, SNIP, OBD, and OBS), which can be categorized into three classes: weight magnitude, linear and quadratic models of the loss function. In the experiments, the authors seek to answer the questions: 1) how well do each criterion preserve the loss; 2) how does the locality assumption affect the final performance; and 3) how does the loss relate to the final performance? Empirical, the authors found that the quadratic model preserves the loss the best, as expected. Also, the loss after pruning seems not strongly correlated with the performance after fine-tuning.

Overall:

This paper is well-written and easy to follow. The authors did a great job of unifying the analysis of several pruning algorithms. More importantly, revisiting the loss modeling of network pruning is interesting, and it might invoke further research efforts in better understanding the pruning techniques developed in the past and also inspire researchers in designing improved pruning algorithms. However, I still have the following questions:

- The authors show that the loss after pruning does not correlate strongly with the accuracy after fine-tuning. In Figure 3, the change in the loss ranges from 0 to 5. Can you show the plot with a smaller range of the change in loss, e.g., 0~0.5? I believe a large change in loss means that the pruning results are very close to random, so the comparisons in this regime may not be meaningful.

- For testing the locality assumption, you introduce an L2 penalty on the changes. To me, this is more like a weighted combination of the original pruning criteria and the magnitude pruning criteria. Why not using some other techniques, such as backtracking line search for determining the pruning ratio at each iteration?

- In equation (5), why do we need to take the absolute value? I think preserving the loss is only meaningful when the network is converged. If the model is not converged, then it would be preferable to prune those weights whose removal will decrease the loss. In this sense, the sign of the loss change should not be ignored.

- The third plot in the second row of Figure 1 shows that OBD with more iterations has a larger change in the loss. Do you have any explanation for this?

Rating:
- I vote for a weak acceptance due to the above reasons. I believe the studied topic in this paper is important and impactful for the pruning community. In the meantime, it would be great if the author can propose some hypotheses on this phenomenon.  To me, preserving the loss is a way to enforce the pruned network to stay close to the original solution, and in this regime, it's easy for the optimization algorithm to find a good enough solution. I will raise my rating if the authors can address my concerns well during the rebuttal period.

==========================After rebuttal=================================

Thanks very much for your efforts to address my concerns. I kept my score unchanged. I agree with most of the responses, except for the response to "L2 penalty, backtracking line search for determining the pruning ratio". This paper is not proposing a practical algorithm but a revisiting, so I don't think the computational cost is a bottleneck in preventing you from using more advanced methods to get more robust conclusions.

---

> ### Author Response · Authors · 2020-11-14
> **Response to Reviewer5**
>
> Thank you for  taking the time to review our paper! We appreciate that you found revisiting loss modelling interesting and that it could inspire researchers in designing improved pruning algorithm.
>
>
> ## Smaller range of the change in loss, e.g., 0~0.5
>
> Thanks for the suggestion. We added a zoomed version of Figure 3 in the Appendix (Figure 14). However, we can not zoom as much as requested in these plots, as the best performing criteria are only able to go as low as 0.6 in training loss at 95.6 % sparsity.
>
> That said, we also added a zoomed version of Figure 11 (Figure 12), which contains scatter plots at 89.3 % sparsity. At that sparsity level, the best criteria can produce pruned networks with 0.06 of training loss, and thus fit the requested range. We observe a similar lack of correlation in these plots.
>
> An interesting related observation from Figure 3 is the fact that there are models producing close to random predictions that still produce extremely good results after fine-tuning, which reinforce even more the argument of not looking solely at loss preservation when designing pruning criteria.
>
>
> ##  L2 penalty, backtracking line search for determining the pruning ratio
>
> The idea of using L2 penalty for the locality assumption is actually coming from Trust Region methods. We selected it for its simplicity and, as you mentioned, for its interesting link with MP. We were interested in investigating the impact of the locality assumption rather than investigating the best method to actually enforce it.
>
> We agree that more advanced methods such as backtracking line search could indeed be used instead. It would come at a higher computational cost, as several extra forward passes need to be performed, but one would not have to tune the extra L2 penalty hyper-parameter, which would be quite convenient and save computations.
>
>
> ## Absolute value in equation (5)?
>
> We agree that removing the absolute value in equation (5) is an interesting direction to explore, as there shouldn’t be any reason to prevent the pruning to produce pruned networks with better loss.
>
> We explored pruning criteria that did not use the absolute value, but we found out that it was extremely hard to prune models without the absolute values. We hypothesize that the main issue with that is that when removing parameters with the smallest (negative) sailencies would correspond to removing parameters that improve the most the loss, but these parameters could also be the ones for which the model is likely to be wrong. However, further research is needed to clearly understand this behaviour.
> We also note that SNIP (Lee et al., 2019), Molchanov et al. (2017) and Molchanov et al. (2019) also use the absolute values in their formulation, although they do not apply the pruning on converged networks either.
>
>
> ## OBD with more iterations has a larger change in the loss. Do you have any explanation for this?
>
> This is a good question. Since the pruning is performed iteratively, the convergence assumption of OBD is violated, so it is surprising that OBD is working that well on VGG11. One hypothesis could be that in the first few stages of pruning, when the network is close to convergence and thus the convergence assumption is quite well respected, then the model is quite faithful and thus the pruning criterion produces good results. However, the more stages are performed, the worse the convergence assumption becomes, and the poorer the pruning becomes. In that case, it would probably be better to prune more parameters at once at the beginning, and that would explain why performing a smaller number of iterations would be more beneficial.
>
>
> ## Some hypotheses explaining the lack of correlation between performance before and after fine-tuning. Preserving the loss is a way to enforce the pruned network to stay close to the original solution, and in this regime, it's easy for the optimization algorithm to find a good enough solution.
>
> We originally shared the same intuition about loss-preserving criteria and this is the original motivation of this work. However, our empirical evaluation highlights that there is a trade-off between function preserving capability and re-trainability: the parameters that preserve best the function might not be the one that ensure the good re-trainability of the network. One possible explanation is that parameters that preserve best the network do not maximize the gradient flow of the networks.
>
> We demonstrated that criteria that only focused on gradient flow such as GraSP (Wang et al., 2020), and not preserving the function, were able to achieve competitive performance after fine-tuning (section 6.2) as a corroborating evidence.
> We believe that future research in pruning criteria should be mindful of both the function function-preserving ability and network re-trainability.
>
>
> Thank you again for your valuable feedback. We hope we addressed your concerns. Please let us know if you have any other questions.

---

### Author Response · Authors · 2020-11-14
**Genreal Response to Reviewers**

Dear reviewers and area chair,

First we would like to thank you for your detailed reviews. We are pleased to hear that the paper was clearly written (R2, R4, R5) and that its empirical investigation is valuable and appreciated (R3).

Some reviewers seem to have appreciated our contributions and empirical investigation (R2, R5), while other reviewers found our paper was missing novelty and insights (R3, R4). We hope our individual responses to these reviews will help shed the light on the key novel insights that our paper is containing.

Finally, we want to point out that we made the following changes to the paper:
- (R2) We added ImageNet results at 90 % sparsity in Appendix C5 and Figure 17.
- (R3) We added plots showing validation accuracy before fine-tuning in Appendix (Figure 6 and 9).
- (R4) We removed the confusing hypothesis about OBS’ performances that were in Appendix A2.
- (R5) We added zoomed versions of the scatter plots (Figures 12 and 14).
- (R2) We added the proposed reference, as well as a couple of others, in the discussion section of the paper.

---

### Author Response · Authors · 2020-11-20
**Awaiting Reviewer's Comments**

Dear reviewers,

We would like to hear back from you in case some points still need clarification, so that we can provide more details before the rebuttal period is over.

Thanks you!

---

### Decision · Program_Chairs · 2021-01-07
**Final Decision**

**Decision:**

Reject

**Comment:**

This paper presents a systematic breakdown and evaluation of several assumptions and algorithmic choices for pruning algorithms. As covered in the reviews, the evaluation and its conclusion offers a timely contribution to the broader community.

In particular, this paper uncovers the observation that precisely modeling the loss (and hence minimizing the drop in loss after pruning) may not in fact yield improvements in pruning. This is an important observation as the community continues to propose new techniques with the justification that their improved performance results from improved loss modeling.

A significant concern on the part of the reviewers is the limited practical prescription offered by the paper. Specifically, the paper does not propose a new algorithm. It also doesn’t necessarily identify why this interesting phenomenon emerges.  For example, to the latter, it doesn't articulate what features of the network or loss landscape is indicative of this property.

Ultimately, the decision for this paper is very challenging given the reviews. Whether or not a phenomena is interesting is an inherently subjective consideration. Moreover, without a clear technical prescription or path forward that can be evaluated on its merits, the reviews fall into two categories of either 1) those that --- by my estimation --- felt personally inspired by the work and 2) those that could not intuit the impact of the observation.

A significant complication is that the narrative of the paper includes claims around addressing locality and convergence which, if not read with the understanding that contributions here are simply a synthesis of current work, appear as claims to novelty (when these techniques have no or limited novelty). This is a source of contention in at least one review.

Given this partition, my recommendation is Reject.

For future versions of this paper, I recommend that the authors narrow the claimed contributions to exclusively focus on the final observation that modeling the loss may not be as important as thought.

The work in this paper on developing the ideas around convergence and locality can, instead, be cast as efforts to provide best available baselines for the topline claim.  I believe these changes will eliminate a significant source of distraction, enabling readers (and reviewers) to avoid any attempt to evaluate the novelty of  the locality and convergence narratives, which have indeed been considered in other work in various ways.

An additional step that I highly recommend for this paper to unambiguously clear the bar is to identify with what the performance of pruning does correlate. Appendix C.4 provides an evaluation of two recent gradient preservation methods. Unfortunately, the paper did not present if, instead, the preservation of the gradient correlated with additional performance.

In essence, the paper need not solve the mystery by providing a SoTA algorithm that exploits the right features of the problem for pruning. However, it would be valuable to provide a roadmap for future directions along with an articulation of the challenges down those directions.